# The Impact of a Post-Prescription Review and Feedback Antimicrobial Stewardship Program in Lebanon

**DOI:** 10.3390/antibiotics11050642

**Published:** 2022-05-11

**Authors:** Anita Shallal, Chloe Lahoud, Dunia Merhej, Sandra Youssef, Jelena Verkler, Linda Kaljee, Tyler Prentiss, Seema Joshi, Marcus Zervos, Madonna Matar

**Affiliations:** 1Division of Infectious Diseases, Henry Ford Hospital, Detroit, MI 48202, USA; ashalla2@hfhs.org (A.S.); jverkle1@hfhs.org (J.V.); sjoshi5@hfhs.org (S.J.); mzervos1@hfhs.org (M.Z.); 2Division of Infectious Diseases, Brigham & Women’s Hospital, Harvard Medical School, Boston, MA 02115, USA; clahoud@bwh.harvard.edu; 3Notre Dame des Secours University Hospital, Byblos 1401, Lebanon; merhejdunia@gmail.com (D.M.); sandra.j.youssef@gmail.com (S.Y.); 4Global Health Initiative, Henry Ford Hospital, Detroit, MI 48202, USA; lkaljee1@hfhs.org (L.K.); tprenti1@hfhs.org (T.P.); 5Office of Global Affairs, Wayne State University School of Medicine, Detroit, MI 48202, USA; 6School of Medicine and Medical Sciences, Holy Spirit University of Kaslik, Byblos 1401, Lebanon

**Keywords:** global health, antimicrobial stewardship, COVID-19, disaster planning

## Abstract

Antimicrobial stewardship programs (ASPs) are effective means to optimize prescribing practices. They are under-utilized in the Middle East where many challenges exist for ASP implementation. We assessed the effectiveness of infectious disease physician-driven post-prescription review and feedback as an ASP in Lebanon. This prospective cohort study was conducted over an 18-month period in the medical, surgical, and intensive care units of a tertiary care hospital. It consisted of three phases: the baseline, intervention, and follow-up. There was a washout period of two months between each phase. Patients aged ≥16 years receiving 48 h of antibiotics were included. During the intervention phase, the AMS team reviewed antimicrobial use within 72 h post-prescription and gave alternate recommendations based on the guidelines for use. The acceptance of the recommendations was measured at 72 h. The primary outcome of the study was days of therapy per 1000 study patient days. A total of 328 patients were recruited in the baseline phase (August–October 2020), 467 patients in the intervention phase (January–June 2021), and 301 patients in the post-intervention phase (September–December 2021). The total days of therapy decreased from 11.46 during the baseline phase to 8.64 during the intervention phase (*p* < 0.001). Intervention acceptance occurred 88.5% of the time. The infectious disease physician-driven implementation of an ASP was successful in reducing antibiotic utilization in an acute care setting in Lebanon.

## 1. Introduction

Antimicrobial resistance (AMR) is an urgent threat to global health; contributing to over 8 million hospital stays, it costs healthcare systems over USD 20 billion [1] and is anticipated to result in the death of 10 million people per year by 2050 [2,3]. AMR is a unique challenge, particularly in areas of the Middle East where health services have been severely impacted by conflict [4]. In these collapsed health systems, there may be a lack of laboratories and other diagnostic tools amidst the unregulated use of antibiotics [4]. In one observational study, multidrug-resistant organisms were frequently detected in water in Lebanon, serving as a potential reservoir for the dissemination of resistant organisms [5]. Furthermore, conflict contributes to a shortage of healthcare providers, further worsening the overall crisis [6]. The lack of overall data and surveillance of AMR in this region makes it especially difficult to compare it with the rest of the world, thus limiting opportunities for intervention. More recently, barriers for the region were summarized and included a need for the training of healthcare personnel and an increased laboratory capacity for the surveillance of AMR as well as the development of antimicrobial stewardship (AMS) guidelines [7].

The most important causes of AMR are inappropriate antimicrobial prescribing and use [2]. Recognizing the urgency of this issue, various AMS programs (ASPs) have been implemented to reduce AMR and promote proper prescribing practices. In order for ASPs to be effective, significant infrastructural components must be present, including national and institutional prescribing policies, healthy prescriber–pharmacist relationships, surveillance and reviews of prescribing practices, and stakeholder buy-ins [8]. Institutions in the Middle East are lacking in firm guidelines for proper antimicrobial use. One systematic review identified 20 studies that described potential proactive interventions with impacts on the de-escalation of antibiotics, discontinuation rates of restricted antibiotics, and length of hospital stay, and that prospective audits and feedback were beneficial for the clinical outcome [1]. Yet, it was also noted that cultural considerations in the Middle East such as physician attitudes, the acceptance of collaborative practices, and the acceptance of pharmacist recommendations may limit effectiveness [1].

The post-prescription review and feedback (PPRF) program has been shown to be effective in the United States and in low- to middle-income countries, including Nepal and India [9,10,11]. This method of ASP has not been demonstrated to be effective in the Middle East and North Africa regions. This project evaluated the impact of a PPRF program at a tertiary hospital in Lebanon.

## 2. Materials and Methods

### 2.1. Study Design and Setting

This prospective cohort study was carried out to determine the feasibility of the implementation of an ASP. The study period consisted of 21 months from April 2020 to December 2021, and was divided into 3 phases: the baseline (3 months), intervention (6 months), and post-intervention (4 months). There was a washout period of 2 months between each phase. The project was a collaboration between the Henry Ford Hospital Division of Infectious Diseases (Detroit, MI, USA) and Notre Dame des Secours University Hospital (Byblos, Lebanon). The Notre Dame des Secours University Hospital is a private, not-for-profit tertiary care teaching hospital with 242 beds and covers a diverse population of medical cases from a wide geographic area. For the present study, the PPRF program was piloted in medicine, surgery, and intensive care unit wards.

In this institution, there was no integrated multidisciplinary AMS team. A team was established to supervise and complete all phases of the project. The dedicated AMS team was composed of an ID specialist, and infection prevention and control officers. The institutional review boards of Notre Dame des Secours University Hospital and Henry Ford Hospital approved the study. A waiver of consent was obtained due to the nature of the study.

### 2.2. Identification of the Study Participants

Participants were selected from medical, surgical, and intensive care unit wards and were managed by a primary treating team who remained the same throughout the study. The inclusion criteria were that all patients were aged ≥16 years and had received antibiotics for ≥48 h. The exclusion criteria were patients aged ≤16 years and on antibiotics for ≤48 h. All medical and surgical specialties were included in the study and patients fulfilling the eligibility criteria were recruited daily. The medical records were reviewed and data were collected at each phase of the study, including the demographic data (age and gender), comorbidity, infection type, antibiotic used, length of hospital stay, and total antibiotic days.

### 2.3. Design of the Phases and the Evaluation of Antimicrobial Use

The baseline phase involved 3 months of evaluation of antimicrobial therapy without the provision of any recommendations. Meanwhile, continuous education was initiated and included regional AMR data and the prevention and management of resistance as well as the components and goals of the ASP, details regarding the PPRF implementation within the institution, and a review of the medical guidelines. Sessions were delivered to physicians in different specialties. Practice guidelines were prepared and supervised by the ID specialist and sent to all physicians working at the institution. The guidelines contained recommendations for an empiric therapy (given for early sepsis with an unclear etiology) as well as a definitive therapy for common infectious syndromes, intravenous to oral conversions, renal-adjusted dosing, and the duration of the antibiotic therapy. Due to the COVID-19 pandemic, webinars were scheduled to educate the healthcare staff and address questions regarding these topics.

The intervention phase occurred following a 2-month washout period. The phase consisted of 6 months of intervention, during which the AMS team initiated a consultation for all eligible patients. During the AMS consultation, the ID physician was responsible for informing the primary treatment team of any recommendations about an antibiotic change. Types of interventions included dose optimization, the escalation of therapy, the de-escalation of therapy, route changes, drug discontinuation, the optimization of administration modalities, and “other”. The final decision to act on the recommendations was left to the primary team. The willingness to accept the intervention, type of primary service, and reasons for not accepting the intervention were also recorded (“not convinced”, “felt insecure”, or “needed a longer duration”). For this study, “not convinced” implied that they were not convinced by the AMS team to accept the recommendation and “felt insecure” implied a lack of confidence in changing antimicrobials.

The final (post-intervention) phase occurred after a 2-month washout period. It was designed to re-assess antibiotic prescriptions in the same way as the baseline phase for a period of 4 months to assess if the ASP could be sustained without an intervention. During each period, AMS education continued via webinars, facility-based guidelines, and emails.

### 2.4. Outcome Measures

The primary outcome measure was the antimicrobial DOT at the baseline, intervention, and post-intervention phases. In this study, patient antibiotic days included the day on which the treatment with the antibiotic began through the stoppage of the drug or hospital discharge. Additionally, information regarding the class/type of antibiotic utilized in each phase and for what illness was also compared. The patient-specific outcome was the length of hospital stay per phase.

At the time of the PPRF launch, an email was sent to all physicians within the hospital introducing the ASP. A survey was prepared and sent by email to all 106 physicians in the institution admitting patients to the 3 mentioned units to evaluate the qualitative data, including the perceptions and attitudes regarding antibiotics, level of basic knowledge about antibiotic prescriptions and resistance, and outcome data that could be used to identify potential barriers and facilitators in the implementation and dissemination of future ASPs.

### 2.5. Statistical Analysis

Continuous data were described using means and standard deviations whereas categorical data were described using counts and column percentages. Univariate two-group comparisons were conducted using independent two-group *t*-tests for the continuous variables and chi-squared tests for the categorical variables. Two Poisson regression models were utilized to assess the magnitude and significance of the intervention phase on antibiotic days and hospitalization treatment days whilst adjusting for the confounders of age, gender, and treatment indication category. The total DOT per 1000 study patient days (PDs) was calculated at the baseline, intervention, and post-intervention periods for each antibiotic. An ANOVA was used to compare the mean antibiotics per day between the phases. The proportion of each variable was compared between the baseline and intervention time points, then between the intervention and post-intervention time points, then between the baseline and post-intervention periods using tests of proportion. These values were then compared using proportion tests with a Bonferroni adjustment between the phases. The statistical significance was set at *p* < 0.05. The analyses were performed using Epi Info-7 (Centers for Disease Control and Prevention, Atlanta, GA, USA) and SAS 9.4 (SAS Institute Inc., Cary, NC, USA).

## 3. Results

### 3.1. Study Population

A total of 328 patients were recruited in the baseline phase (August–October 2020), 467 patients in the intervention phase (January–June 2021), and 301 patients in the post-intervention phase (September–December 2021) (Table 1). There was an even distribution of gender across the baseline and intervention, but significantly more females in the post-intervention period (*p* = 0.045). The mean age of patients was 67.6 years in the baseline group; patients were significantly younger in the intervention group compared with pre-intervention, which occurred in a COVID-19 surge (*p* = 0.002). In the baseline period, compared with both the intervention and post-intervention periods, there were significantly more patients with pulmonary comorbidities (32% vs. 6.2% vs. 6.9%, *p* < 0.001). There were significantly more patients with a hepatic/gastrointestinal comorbidity in the intervention and post-intervention periods when compared with the baseline period (1.5% vs. 4.9% vs. 4.7%, *p* = 0.01). Oncologic and renal diseases were significantly more common in the post-intervention period (17.8%, *p* = 0.001 and 14.5%, *p* = 0.002, respectively).

### 3.2. Evaluation of Antimicrobial Use

The type of infection treated varied considerably between the phases. However, across all 3 phases, the most common infection treated was pneumonia. In the intervention phase, there was a significantly higher number of patients receiving an empirical treatment (17.5% compared with 3.1% at the baseline, *p* < 0.001); this was also true for the post-intervention phase (7.6%, *p* < 0.001). The number of patients treated for a gastrointestinal infection or a urinary tract infection and receiving postoperative prophylaxis significantly decreased in the intervention and post-intervention phases and was most significant between the baseline and intervention phases (13.2% vs. 3.8%, *p* < 0.001; 25.1% vs. 13.5%, *p* < 0.001; and 11.1% vs. 4.1%, *p* < 0.001, respectively). Skin and soft tissue infections were significantly more common in the post-intervention phase (11.1% vs. 2.6% in intervention, *p* < 0.004). In the intervention phase, which occurred during a COVID-19 surge, empiric therapies (3.1% vs. 17.5%, *p* < 0.001) and “other” infections (9.45% vs. 31.8%, *p* < 0.001) significantly increased (Table 2).

The choice of antimicrobial agent also significantly differed between the phases. There was an increase in days of vancomycin use compared with the baseline (57.83 DOT/1000 PD vs. 101.03, *p* < 0.01), with higher results in the post-intervention period compared with the baseline (150.26, *p* < 0.01). There was a significant reduction in the use of carbapenems between the baseline and intervention periods (455.62 DOT/1000 PD vs. 322.75, *p* < 0.01) with an increase in post-intervention prescribing in both phases (567.02 DOT/1000 PD, *p* < 0.01, *p* < 0.01). Beta-lactam use increased during the intervention phase (139.23 DOT/1000 PD to 187.24, *p* < 0.01) and returned to similar levels during post-intervention (122.97 DOT/1000 PD, *p* < 0.01) when compared with the intervention phase. In the post-intervention period, an increase in cephalosporin and carbapenem use was noted when compared with the baseline period; however, this did not reach a statistical significance. There was also a significant increase in the duration of aminoglycoside use in the post-intervention period when compared with the intervention period (4.6 vs. 2.9, *p* = 0.004) (Table 3).

### 3.3. Intervention Acceptance

In the intervention phase, there were 467 interventions from the AMS team, which were accepted 414 times (88.5%). The most commonly performed intervention was “other” (234 (56.2%)), followed by drug discontinuation (108 (25.9%)). The escalation of therapy and the de-escalation of therapy occurred 25 times, respectively (6.1%, respectively). Dose optimization, route change, and administration modality optimization were less frequently performed (17 (4%), 5 (1.2%), and 2 (0.5%), respectively). For the 54 occurrences where an intervention was not accepted (11.5%), the reason for not accepting was most commonly due to a report of being “not convinced” (41 (75.9%)). Feeling insecure and needing a longer duration of therapy were less common reasons of not accepting (5 (9%) and 9 (16%), respectively). Among the types of interventions, the intervention that was most commonly not accepted was drug discontinuation (24%). Among the treatment providers, the primary service that did not accept an intervention the most frequently was surgery (15.9%) compared with medicine (12.4%) and the intensive care unit (12%).

### 3.4. Clinical Outcomes

Despite the increase in antimicrobial prescriptions for “other” infections and empirical therapies, the total days of antibiotic therapy decreased from 11.46 during the baseline phase to 8.64 days during the intervention phase (*p* < 0.00001). In the post-intervention phase, the total antibiotic days of therapy (DOT) was 10.9, and, although this number was lower than in the baseline phase, the effect did not meet a statistical significance (*p* = 0.57) (Table 1). After adjusting for age, sex, and treatment indication, members in the pre-intervention phase were on antibiotics 29% longer (*p* < 0.001) and hospitalized 16% longer (*p* < 0.001) than members in the intervention phase. After adjusting for age, sex, and treatment indication, members in the pre-intervention phase were on antibiotics 6% longer (*p* = 0.02) and hospitalized 3% longer (*p* = 0.33) than members in the post-intervention phase.

Of the 106 physicians, 20 completed the post-intervention survey, which was a response rate of 18.8%. The majority of respondents (12 (60%)) were female and the median age was 29.6 years old. Most respondents were practicing in the field of medicine (n = 14) and had a median of 6 years since the completion of their highest education. Their perceptions on antibiotic prescribing and knowledge were then assessed. The majority of respondents agreed or strongly agreed that inappropriate antibiotic prescribing put patients at risk and that AMR was a problem in the hospital where they worked. The majority of respondents felt that they had received inadequate training in antibiotic prescribing and that there was a need to increase the education of healthcare providers on AMR and antimicrobial prescribing. The majority of respondents felt that the PPRF would increase AMS without being disruptive to their work. Most respondents noted that they had never prescribed antibiotics because a patient insisted on it or to improve relationships with their patients. The most useful means of increasing antimicrobial prescribing education were felt to be short written guidelines, seminars, and summary written materials (Appendix A).

## 4. Discussion

### 4.1. PPRF Programs

This study was the first in Lebanon to examine the impact of the implementation of an infectious disease (ID) physician-driven PPRF strategy of AMS. In the intervention period, there was a significant reduction in DOT, type of illness treated, and types of antimicrobials in use and an indirect decrease in the length of hospital stay. The PPRF was noted to have many advantages. The program engaged the treating physicians with medical discussions, promoting an opportunity for education within medical teams. It also had a positive impact on collaborative clinical decision-making regarding antibiotic therapy modifications or discontinuations. Studies have shown that PPRFs build trust and rapport between treatment teams [11] with an associated decrease in DOT [12]. In addition, there is an opportunity for the AMS team to evaluate the barriers and enablers that can be used to modify the ASP [13]. At this institution, the PPRF was shown to be effective by not only decreasing the total DOT, but also by reducing the use of broad-spectrum antimicrobials such as carbapenems and beta-lactam/beta-lactamase inhibitors. The ASP had an indirect impact on the length of hospital stay of the patients, which was shorter in the intervention phase after adjusting for age, sex, and the treatment indication. Given that COVID-19 pandemic waves occurred in that time period, this could have been a bias that was not accounted for. However, this reduced length of hospital stay as a result of the ASP has also been noted in other studies [14].

Notably, the acceptance of the AMS team recommendations was 88%, which was higher than in prior studies that typically noted an acceptance rate of 60–70% [11,15,16]. We believed that the type of patients admitted (the majority was pneumonia, particularly given the COVID-19 surge in the intervention phase) and the use of verbal feedback and discussions as well as the continued stream of educational materials throughout the three phases of the study may have had an impact on the acceptance of recommendations. Additionally, the intervention was ID physician-led rather than pharmacist-led, as prior studies in the Middle East have noted less acceptance from pharmacists [15]. The most frequent reason for not accepting recommendations was the feeling of being “not convinced”, which implied that there may be further work to be undertaken with further education on antibiotics with teams. Importantly, it is worth noting that feeling “not convinced” may actually relate to a need for behavioral change, with less reliance on personal knowledge or experience and more reliance on guidelines and formal policy [16,17]. Acceptance rates in this study were notably lower among surgical specialties, which was consistent with a prior similar study in Nepal [9]. Management guidelines across different specialties could strengthen the treating of the certainty of the clinical decisions of physicians, particularly when there are unified treatment algorithms according to local epidemiology data [18]. Furthermore, the integration of rapid diagnostic tools and the strengthening of laboratory capacities could help clinicians in their decision-making when empirically treating complicated cases [19].

### 4.2. Physician Attitudes

This study took into account the attitudes toward the ASP of the treating physicians at the study hospital. Although the number of participants in the survey was low, there was a general consensus among the participants about perceptions and barriers toward a PPRF. Additionally, the survey responses alerted a potential lack of confidence in knowledge about antibiotic prescribing and issues with insufficient training. The need for regular educational programs on prescribing has been cited in other ASPs [20]. Potential options for educational programs could include workshops and guidelines; prescribers tend to prefer guidelines locally developed rather than nationally developed [16]. Continued medical education should include seminars and modules with case studies and these educational programs should be routinely audited to ensure they can be integrated into prescriber schedules [9]. Importantly, three participants in the survey felt that the PPRF could be disruptive to the patient treatment. A recent study, also in Lebanon, evaluated physician attitudes toward the core elements of an ASP and found that 34% of participants felt that the ASP could affect physician autonomy [21]. Less restrictive stewardship programs, including PPRFs and antibiotic rounds, are possible ways to circumvent this issue, which may stem from cultural aspects of medicine in the region [15,22].

### 4.3. Impact of the Timeline on the Study

The study timeline directly coincided with COVID-19 surges that impacted on each phase. Two-month washouts were performed due to delays from the impact of COVID-19 and the economic collapse. Due to the increased hospitalization rate of COVID-19 patients, many were admitted to floors that were part of the intervention phase of the study and thus included in the PPRF. Admitted COVID-19 patients were frequently empirically treated for pneumonia, providing a unique opportunity for the AMS team to intervene, given prior studies that showed that although more than 90% of hospitalized COVID-19 patients received empiric antibiotics, only 15% were documented to have a secondary bacterial infection [23,24]. The rates were lower in other observational reports, increasing the hospital stay from 3.2% up to 6.1% [25]. However, it was unclear whether the COVID-19 surges in this study also led to greater antibiotic de-escalation as well as more readily acceptable interventions from our AMS team as a result of the known (viral-etiology) pandemic.

The economic collapse of Lebanon—which began in October 2019, but has significantly progressed since then (6)—directly impacted on the post-intervention phase of this study. During the post-intervention phase, there was a severe shortage of many antibiotics. Antibiotics were thus selected based on availability and not per empiric or treatment guidelines. This was clearly noted in our results where the use of carbapenems, cephalosporins, aminoglycosides, and colistin were higher in the post-intervention period. During this phase, an emphasis on the duration of therapy rather than the antibiotic agent choice was placed within stewardship education.

### 4.4. Strengths and Limitations

Separate from the direct impact of the economic collapse and COVID-19, there were other limitations to our study. Logistic challenges in place resulted in shortages of providers, which may have affected the ease of education of the treatment teams. Additionally, we did not take into account the severity of disease, mortality/survival benefit, or types of multidrug-resistant organisms treated, which might have impacted on the acceptance rate of the interventions, the type of interventions, and the type of antibiotics used. Finally, due to delays, the study was limited to an 18-month period, making assessments of changes to the AMR and susceptibility patterns impossible. This was also compounded by factors outside the control of a hospital program such as outpatient antimicrobials and antibiotic use in agricultural feeds.

A number of strengths exist in this study. First, the large number of patients recruited helped power the study appropriately for the analysis. Furthermore, the interrupted time series with washout periods between the phases reduced any bias that may have been associated with recent and direct education provided by the AMS team. Finally, the interventions were made with active dialogue and constant interaction with the treatment teams. This may have accounted for the increased acceptance of the recommendations compared with those PPRFs that use electronic medical record alerts. Using multidisciplinary collaboration and education were essential components of the ASP. Although labor-intensive, the overall primary outcome was clear and the secondary impact on the cost analysis given the reduction in the length of hospital stay is a potential direction for future studies.

## 5. Conclusions

Multifaceted approaches are needed to combat AMR across the Middle East. AMS education and strategies, including PPRFs, are useful methods to reduce DOT.

## Figures and Tables

**Table 1 antibiotics-11-00642-t001:** Summary of demographic data and days of therapy.

Demographics	Pre-Intervention (*n* = 328)	Intervention (*n* = 467)	Post-Intervention (*n* = 301)	*p*-Value ^a^ Phase 1 vs. 2 ^b^ Phase 1 vs. 3 ^c^ Phase 2 vs. 3
Gender, female	40.24% (132)	38.89 (182)	46.48% (99)	0.69 ^a^, 0.05 ^b^, 0.08 ^c^
Mean age (SD)	67.6 (16.7) Range 17–97	63.72 (17.65) Range 16–96	65.84 (17.4) Range 16–97	<0.01 ^a^, 0.02 ^b^, 0.10 ^c^
Median LOS (SD)	6.00 (6.7) Range 1–41	6.00 (4.8) Range 1–42	7.00 (5.1) Range 3–75	<0.01
Median duration of antibiotic course	5.00 (6.27) Range 1–40	5.00 (4.26) Range 1–39	6.00 (5.01) Range 2–25	<0.01
Median duration of antibiotic days/patient (SD)	8.00 (12.44) Range 1–118	7.00 (7.4) Range 1–63	8.0 (9.46) Range 1–63	<0.01
Pulmonary	32.01% (105)	6.21% (29)	6.9% (21)	<0.01 ^a^, <0.01 ^b^, 0.70 ^c^
Cardiac	51.83% (170)	52.68% (246)	48.7% (148)	0.81 ^a^, 0.43 ^b^, 0.29 ^c^
Vascular	3.05% (10)	1.93% (9)	2.6% (8)	0.31 ^a^, 0.74 ^b^, 0.50 ^c^
Endocrine	35.98% (118)	39.4% (184)	35.5% (108)	0.33 ^a^, 0.90 ^b^, 0.29 ^c^
Neurologic	5.79% (19)	7.49% (35)	4.6% (14)	0.35 ^a^, 0.50 ^b^, 0.11 ^c^
Hepatic/GI	1.52% (5)	4.93% (23)	3.9% (12)	0.01 ^a^, 0.06 ^b^, 0.53 ^c^
Heme/onc	9.15% (30)	14.99% (70)	17.8% (54)	0.01 ^a^, <0.01 ^b^, 0.29 ^c^
Renal	7.01% (23)	8.57% (40)	14.47% (44)	0.42 ^a^, <0.01 ^b^, 0.01 ^c^
Other	6.71% (22)	16.9% (79)	19.1% (58)	<0.01 ^a^, <0.01 ^b^, 0.43 ^c^

**Table 2 antibiotics-11-00642-t002:** Type of infection treated in each phase.

Treatment Indication	Pre-Intervention (*n* = 326)	Intervention (*n* = 467)	Post-Intervention (*n* = 301)	*p*-Value ^a^ Phase 1 vs. 2 ^b^ Phase 1 vs. 3 ^c^ Phase 2 vs. 3
Empirical treatment	3.07% (10)	17.5% (82)	7.6% (23)	<0.001 ^a^, 0.01 ^b^, 0.001 ^c^
Pneumonia	26.69% (87)	23.3% (109)	22% (67)	0.276 ^a^, 0.17 ^b^ 0.69 ^c^
Gastrointestinal	13.19% (43)	3.85% (18)	3.9% (12)	<0.001 ^a^, <0.001 ^b^ 0.94 ^c^
Sepsis	3.05% (10)	1.92% (9)	2.3% (7)	0.298 ^a^, 0.55 ^b^, 0.71 ^c^
Urinary tract infection	25.15% (82)	13.46% (63)	18.1% (55)	<0.001 ^a^, 0.03 ^b^, 0.08 ^c^
Postoperative prophylaxis	11.04% (36)	4.06% (19)	2.9% (9)	0.001 ^a^, 0.001 ^b^, 0.42 ^c^
Skin and tissue infection	4.91% (16)	2.56% (12)	11.1% (34)	0.0784 ^a^, 0.004 ^b^, <0.0001 ^c^
Diabetic foot infection	3.37% (11)	1.5% (7)	1.3% (4)	0.080 ^a^, 0.09 ^b^, 0.83 ^c^
Other	9.45% (31)	31.8% (149)	32.9% (100)	<0.001 ^a^, <0.001 ^b^, 0.76 ^c^

**Table 3 antibiotics-11-00642-t003:** Antibiotic agent used in each phase.

DOT/1000 by Agent Mean	Baseline (*n* = 328, Patient Days 2715)	Intervention (*n* = 467, Patient Days 3306)	Post-Intervention (*n* = 301, Patient Days 2529)	*p*-Value ^a^ Phase 1 vs. 2 ^b^ Phase 1 vs. 3 ^c^ Phase 2 vs. 3
Intravenous antibiotics	1317.86 1250.18	1205.08 1197.96	1296.56 1263.3	0.82
Oral antibiotics	67.03	18.15	25.32	<0.01 ^a^, <0.01 ^b^, 0.06 ^c^
Vancomycin	57.83	101.03	150.26	<0.01 ^a^, <0.01 ^b^, <0.01 ^c^
Linezolid	15.84	11.8	22.14	0.53 ^a^, 0.28 ^b^, <0.01 ^c^
Trimethoprim-sulfamethoxazole	11.42	0	0	<0.01 ^a^, <0.01 ^b^
Doxycycline	7	0	1.98	<0.01 ^a^, 0.02 ^b^, 0.03 ^c^
Penicillin	6.26	2.12	17	0.03 ^a^, < 0.01 ^b^, <0.01 ^c^
Beta-lactam/BLI	139.23	187.24	122.97	<0.01 ^a^, 0.25 ^b^, <0.01 ^c^
Cephalosporin	306.08	399.58	248.71	<0.01 ^a^, <0.01 ^b^, <0.01 ^c^
Carbapenem	455.62	322.75	567.02	<0.01 ^a^, <0.01 ^b^, <0.01 ^c^
Metronidazole	60.77	51.72	27.68	0.38 ^a^, <0.01 ^b^, <0.01 ^c^
Azithromycin	2.94	9.68	5.54	0.04 ^a^, 0.42 ^b^, 0.22 ^c^
Clindamycin	2.58	9.98	17	<0.01 ^a^, <0.01 ^b^, 0.06 ^c^
Fluoroquinolone	95.03	88.02	67.62	1.000 ^a^, <0.01 ^b^, 0.01 ^c^
Colistin	0	0	25.3	<0.01 ^b^, <0.01 ^c^
Aminoglycoside	20.99	11.19	16.2	0.01 ^a^, 0.6 ^b^, 0.29 ^c^
Tigecycline	13.26	25.11	18.2	<0.01 ^a^, 0.45 ^b^, 0.23 ^c^
Other	189.3	0	9.49	<0.01 ^a^, <0.01 ^b^, <0.01 ^c^

BLI: beta-lactamase inhibitor; SD: standard deviation.

## Data Availability

Not applicable.

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
