# Peer review of "The Impact of a Post-Prescription Review and Feedback Antimicrobial Stewardship Program in Lebanon"

_antibiotics, 2022, doi:10.3390/antibiotics11050642_

Round 1

Reviewer 1 Report

This was an interesting and well designed study. I would like to commend the Authors on their work. Considering LMICs are expected to bear the highest burden of AMR, data on effective strategies to curb AMR development and spread in this context are important. I suggest some revisions.

Introduction:

l.45 was -> were

l. 50 what does "key barriers" refer to? 

l. 56 proper -> appropriate

Methods:

I find the placement of this section (after discussion) confusing.

l. 329 Statistical analysis should be reassessed. Two groups are compared, when in fact there are three groups.

l. 332-335 this section is unclear.

Results:

All tables: how were data compared, what do p-values refer to? Please revise the text accordingly. Please specify what values refer to (numbers, percentages, ect) and provide units.

Table 1: Illness -> does this refer to comorbidities or treatment indication?

Table 2: infection treated -> treatment indication

Table 3: add total row

l. 138-140: there is no mention of this analysis in methods section.

Tables 5-7: should be supplementary material, as main results are reported in text. Please add %.

Discussion:

l. 171 aand 181-182: Please elaborate on the indirect effect on LOS. The study design does not allow to make any assumptions on a causal relationship nor on the direction of association, and several biases could have affected the relation between these two variables.

l. 179-180: the increased use of colistin is not discussed. Could this be due to shortages in other antibiotics (l. 234-235)?

l. 185-219: the assessment of willingness to accept the intervention is an interesting addition to this study. However, it would have been interesting to further discuss restrictive AMS strategies in light of the cultural aspects highlighted in this study (l. 218-219).

l. 251: regency bias?

Author Response

This was an interesting and well designed study. I would like to commend the Authors on their work. Considering LMICs are expected to bear the highest burden of AMR, data on effective strategies to curb AMR development and spread in this context are important. I suggest some revisions. We thank the reviewer for their comments.

Introduction:

l.45 was -> were. Done

l. 50 what does "key barriers" refer to? Edited to remove “key”

l. 56 proper -> appropriate Done

Methods:

I find the placement of this section (after discussion) confusing. This was done due to the manuscript guidelines posted. We have edited and replaced it to before results.

l. 329 Statistical analysis should be reassessed. Two groups are compared, when in fact there are three groups. All 3 groups are compared using separate z-tests.

l. 332-335 this section is unclear. Each group was compared to each other to assess directionality. One chi-squared test of independence would not provide information on directionality.

Results:

All tables: how were data compared, what do p-values refer to? Please revise the text accordingly. Please specify what values refer to (numbers, percentages, ect) and provide units. The 1st p-value was pre vs intervention, 2nd p-value was intervention vs post-intervention, and 3rd p-value was pre and post intervention.

Table 1: categorical variables are z-test and continuous variables are t-tests. The units are specified in the row name.

Table 2: all z-tests of proportion and counts.

Table 3: all t-tests and days as the unit.

Table 1: Illness -> does this refer to comorbidities or treatment indication? Edited table. This referred to co-morbid conditions.

Table 2: infection treated -> treatment indication Edited. This referred to treatment indications.

Table 3: add total row A total row would imply antibiotic days, which is supplied in table 1. Table 3 was completed re-analyzed as per reviewer 2 suggestions.

l. 138-140: there is no mention of this analysis in methods section. Added to methods: Two Poisson regression models were utilized to assess magnitude and significance of intervention phase on antibiotic days and hospitalization treatment days while adjusting for the confounders of age, gender, and treatment indication category.

Added to results: after adjusting for age, sex, and treatment indication, members in the pre-intervention phase were on antibiotics 29% longer (p<0.001) and hospitalized 16% longer (p<0.001) than members in the intervention phase. After adjusting for age, sex, and treatment indication, members in the pre-intervention phase were on antibiotics 6% longer (p=0.02) and hospitalized 3% longer (p=0.33) than members in the post-intervention phase.

Tables 5-7: should be supplementary material, as main results are reported in text. Please add %. Moved to after references as supplementary data.

Discussion:

l. 171 aand 181-182: Please elaborate on the indirect effect on LOS. The study design does not allow to make any assumptions on a causal relationship nor on the direction of association, and several biases could have affected the relation between these two variables. We added a sentence about this and included that the pandemic may have affected hospitalization LOS as well.

l. 179-180: the increased use of colistin is not discussed. Could this be due to shortages in other antibiotics (l. 234-235)? Added in a sentence at the impact of timeline paragraph

l. 185-219: the assessment of willingness to accept the intervention is an interesting addition to this study. However, it would have been interesting to further discuss restrictive AMS strategies in light of the cultural aspects highlighted in this study (l. 218-219). Emphasized.

l. 251: regency bias? Clarified

Reviewer 2 Report

The article describes the impact of a post-prescription review and feedback anti-microbial stewardship program in Lebanon, performed in a difficult time (during COVID-19 surges). Still it seems to be performed quite well, though there are some remarks to be made. Especially the tables do need attention.

Line 45: "multidrug resistant organisms was" should be "multidrug resistant organisms were" 

Table 1: I would suggest to leave the p-values out of the table.

Table 1: Given the big spread of length of hospital stay and days of antibiotic treatment a normal distribution of data seems not plausible. Therefore, it might be better to report a median.

Line 96;“and postoperative prophylaxis”: Should not this be “and receiving postoperative prophylaxis”

Table 2: it is not entirely clear what is meant with empirical treatment. Please explain in the methods

The table could be a bit more neat: percentages are displayed with different number of digits and it is not clear which values are compared in the p-values.

Table 3: here again the p-values are not intuitively.

Line 134: "the total days of antibiotic therapy decreased from 11.46 during the baseline 134 phase to 8.64 days during the intervention phase": Is this the mean total days? Or days of therapy per 1,000 study patient days? 

I could not find the study patient days in the paper, please add those as well. Just mean days of treatment do not seem to be the best way to report the results.

Line 138: “After adjusting for age, sex, and primary disease treated, members in the base- 138 line period were on antibiotics for 29% longer and hospitalized for 16% longer than members in the intervention period.” Not sure what analysis was exactly performed here. There is also no information on this in the methods.

Table 4: extra information could be added to the table, like a range for age, years since completion of higherst education, years in currect position or at current hospital.

What was the response rate on the survey?

Tables 5, 6 and 7: numbers do not always add op to 20. Please check

Discussion: could COVID-19 in itself also have led to less antibiotic interventions and shorter duration of treatment? It would be good to elaborate on  this in the discussion

Line 307: to me it is not entirely clear what is the difference between “not convinced, felt unsecure, or needed longer duration” as it seems to be overlapping answers?

Line 328: would advise to report medians if necessay

Author Response

Line 45: "multidrug resistant organisms was" should be "multidrug resistant organisms were" Edited

Table 1: I would suggest to leave the p-values out of the table.

Table 1: Given the big spread of length of hospital stay and days of antibiotic treatment a normal distribution of data seems not plausible. Therefore, it might be better to report a median. Added

Line 96;“and postoperative prophylaxis”: Should not this be “and receiving postoperative prophylaxis” Edited

Table 2: it is not entirely clear what is meant with empirical treatment. Please explain in the methods. Done.

The table could be a bit more neat: percentages are displayed with different number of digits and it is not clear which values are compared in the p-values. The p values were rounded to the tenth place.

Table 3: here again the p-values are not intuitively. Specify p-values as shown above.

Line 134: "the total days of antibiotic therapy decreased from 11.46 during the baseline 134 phase to 8.64 days during the intervention phase": Is this the mean total days? Or days of therapy per 1,000 study patient days?  I could not find the study patient days in the paper, please add those as well. Just mean days of treatment do not seem to be the best way to report the results. Please note a new, separate analysis for days of therapy was included in the paper with a new table #3.

Line 138: “After adjusting for age, sex, and primary disease treated, members in the base- 138 line period were on antibiotics for 29% longer and hospitalized for 16% longer than members in the intervention period.” Not sure what analysis was exactly performed here. There is also no information on this in the methods. This was added as above.

Table 4: extra information could be added to the table, like a range for age, years since completion of higherst education, years in currect position or at current hospital. Please note the table was changed to supplementary data as per reviewer 1 recommendations.

What was the response rate on the survey? Please note that we added a sentence on response rate, as the survey was sent to 106 participants.

Tables 5, 6 and 7: numbers do not always add op to 20. Please check. Not all questions were answered by each participant.

Discussion: could COVID-19 in itself also have led to less antibiotic interventions and shorter duration of treatment? It would be good to elaborate on  this in the discussion Done.

Line 307: to me it is not entirely clear what is the difference between “not convinced, felt unsecure, or needed longer duration” as it seems to be overlapping answers? To us, not convinced meant that they did not agree with the change in antibiotic. This is different than needing longer duration or feeling insecure, and this is emphasized in the discussion.

Line 328: would advise to report medians if necessary Noted as above.

Round 2

Reviewer 2 Report

The manuscript had improved after the revision, though still some remarks have to be made. 

Unfortunately, presentation of p-values in the tables 1 and 2 are still not intuitively. It is not entirely clear were they refer to.

Besides, I would advice to delete the reported mean LOS and duration of AB treatment from Table 1.

Although it is stated that you elaborated on this; Discussion: could COVID-19 in itself also have led to less antibiotic interventions and shorter duration of treatment? It would be good to elaborate on this in the discussion. It not seems to be added? The added phrases only seem to be on length of hospital stay.

Still, the advice would be to add ranges to Table 1 in the Supplementart materials

Also for Table 3 it is not clear what the displayed numbers refer to. There is a mention of an SD, but I cannot find it in the tables.

Lastly you might want to elaborate on  differences between “not convinced, felt unsecure, or needed longer duration” in the methods.

Author Response

The manuscript had improved after the revision, though still some remarks have to be made. We thank the reviewer for their comments.

Unfortunately, presentation of p-values in the tables 1 and 2 are still not intuitively. It is not entirely clear were they refer to. Adjusted table 1 and 2 so that p values are represented like they are in table 3. We think this better makes the p values clear.

Besides, I would advice to delete the reported mean LOS and duration of AB treatment from Table 1. Removed.

Although it is stated that you elaborated on this; Discussion: could COVID-19 in itself also have led to less antibiotic interventions and shorter duration of treatment? It would be good to elaborate on this in the discussion. It not seems to be added? The added phrases only seem to be on length of hospital stay. Please note that this is mentioned in the first paragraph of the discussion, and again under the “impact of timeline on study” portion of the manuscript.

Still, the advice would be to add ranges to Table 1 in the Supplementart materials. Added into S1.

Also for Table 3 it is not clear what the displayed numbers refer to. There is a mention of an SD, but I cannot find it in the tables. This was an error on our part. There are not SDs to report.

Lastly you might want to elaborate on differences between “not convinced, felt unsecure, or needed longer duration” in the methods. Added.